# Poisoning from Ingestion of Fungus-Infected Cicada Nymphs: Characteristics and Clinical Outcomes of Patients in Thailand

**DOI:** 10.3390/toxins16010022

**Published:** 2023-12-31

**Authors:** Satariya Trakulsrichai, Nattapon Satsue, Phantakan Tansuwannarat, Jatupon Krongvorakul, Jetjamnong Sueajai, Pitak Santanirand, Winai Wananukul

**Affiliations:** 1Department of Emergency Medicine, Faculty of Medicine Ramathibodi Hospital, Mahidol University, Bangkok 10400, Thailand; 2Ramathibodi Poison Center, Faculty of Medicine Ramathibodi Hospital, Mahidol University, Bangkok 10400, Thailand; pum-kps@hotmail.com (N.S.); phantakan.tans@gmail.com (P.T.); winai.wan@mahidol.edu (W.W.); 3Department of Medicine, Faculty of Medicine Ramathibodi Hospital, Mahidol University, Bangkok 10400, Thailand; 4Chakri Naruebodindra Medical Institute, Faculty of Medicine Ramathibodi Hospital, Mahidol University, Samut Prakan 10540, Thailand; 5Department of Pathology, Faculty of Medicine Ramathibodi Hospital, Mahidol University, Bangkok 10400, Thailand; jatupon.krn@mahidol.ac.th (J.K.); jetjamnong.sue@mahidol.ac.th (J.S.); pitak.san@mahidol.ac.th (P.S.)

**Keywords:** cicada nymphs, *Cordyceps* fungus, ibotenic acid

## Abstract

The current data regarding poisoning associated with ingestion of fungus-infected cicada nymphs are limited. We performed a retrospective cohort study of patients who ingested fungus-infected cicada nymphs and were referred to the Ramathibodi Poison Center for consultation from June 2010 to June 2022. Thirty-nine patients were included for analysis. Most were men (53.8%). Mean age was 40.2 ± 15.0 years. All nymphs were ingested as a health/food supplement. Thirty-one patients (79.5%) reported gastrointestinal symptoms. Median time from ingestion to symptom onset was 5 h. Twenty-nine patients (74.4%) reported neurological symptoms, including tremor, myoclonus, muscle rigidity, nystagmus/ocular clonus, drowsiness, dysarthria, seizure, and confusion. Some complained of dizziness, urinary retention, and jaw stiffness. Most patients (94.9%) were admitted to the hospital. Median hospital stay was 3 days. Ibotenic acid was detected in the blood and urine samples of one patient. All received supportive care. Four patients developed infectious complications. No deaths occurred. Consuming fungus-infected cicada nymphs may cause poisoning in humans. Gastrointestinal and neurological symptoms were common. Ibotenic acid might be the underlying cause. The main treatment is supportive care and appropriate management of complications. Education of the general public is advocated to prevent the incidence of this type of poisoning.

## 1. Introduction

Fungi of the genus *Cordyceps* have been used for many years in traditional Chinese medicine (TCM) as a health/food supplement and treatment [1]. Among the more than 700 known species within the genus, *C. sinensis* and *C. militaris* (L.) Fr. are the two most commonly used in herbal medicine [1]. *Cordyceps* species contain numerous chemical compounds, such as cordycepin, and are found in North America, Europe, and numerous countries in Asia, including China, Nepal, Vietnam, and Thailand [1,2].

*Cordyceps* or *Ophiocordyceps sobolifera* is a complex parasitic fungus that grows inside and infects certain species of cicada nymphs and larvae. It is known as chan hua in Chinese, or “cicada flower”. The fungus grows throughout the nymph and destroys it, forming fruiting bodies that sprout from the nymph’s head [2,3]. It has been used as a TCM ingredient for many years [2,3].

In a 2017 study of 60 cases of human poisoning related to ingestion of *Cordyceps* fungus-infected cicada nymphs in Vietnam, taxonomic and chemical analysis of the specimens demonstrated *Ophiocordyceps heteropoda* and the presence of ibotenic acid; muscimol was not detected [2]. The authors proposed that ibotenic acid poisoning was responsible for the clinical poisoning syndrome. Ibotenic acid is the main active substance of *Amanita muscaria* and *Amanita pantherine*, causing poisoning [4,5]. The structure of ibotenic acid resembles glutamic acid and activates N-methyl-D-aspartic acid (NMDA) receptors in the brain resulting in neurotoxicity, especially excitatory effects [6,7].

In Thailand, entomopathogenic fungi that infect cicada nymphs are used in worship and for consumption as an unregistered non-commercial health supplement [8,9]. The infected nymphs are intentionally foraged from soil and collected for consumption. Data regarding human poisoning related to ingestion of fungus-infected cicada nymphs is limited. This study aimed to examine the clinical characteristics and outcomes in patients who experienced this type of poisoning in Thailand over a 12-year period.

## 2. Results

Thirty-nine patients from 23 telephone consultations to the Ramathibodi Poison Center (RPC) were included for analysis. The highest number of patients was reported in 2014 (13 patients), followed by 2018 (7 patients), 2016 (6 patients), and 2021 (5 patients). 

In 2019 and 2022 (half year), there were two patients each, while in 2011, 2013, 2017 and 2020, there was only one patient each. 

Some patients presented to the hospitals together with their family members or friends who had the symptoms after consuming the fungus-infected cicada nymphs; these were defined as poisoning groups. Among patients who presented as poisoning groups, two patients did not develop any symptoms after exposure. 

The patient characteristics are summarized in Table 1. Twenty-one patients were men (53.8%) and 18 were women (46.2%). Mean age was 40.2 ± 15.0 years (range, 10–77 years). Two patients were younger than age 15 years (10 and 13 years, respectively). Thirty-seven patients were Thai; the remaining two were Asian but no specific nationality was recorded.

All patients ingested fungus-infected cicada nymphs as a health/food supplement. Thirty (76.9%) cooked them before eating by frying (25 patients) or boiling (5 patients); three patients did not cook them. Cooking details were not known in six patients. Seven patients ingested nymphs with alcoholic beverages. Two reported underlying disease (hypertension and blindness in both eyes, respectively). 

Most consultations were from the northeast region (43.6%) and Bangkok (33.3%). Figure 1 shows the number of poisoning events over the 12-year study period according to month. Most cases were reported in May, followed by June. We classified the seasons roughly into three 4-month periods: summer (February–May), rainy season (June–September), and winter (October–January) [10]. The majority of cases occurred during the summer season (Table 1). 

The reported amount/quantity of fungus-infected cicada nymphs ingested was difficult to estimate and varied among patients. All patients visited a hospital within 1 day of ingestion. The median time from ingestion to symptom onset was 5 h (range, 0.25–15). The clinical manifestations at presentation are shown in Table 2. Thirty-one patients (79.5%) reported gastrointestinal (GI) symptoms, including nausea, vomiting, and abdominal pain. Twenty-nine patients (74.4%) reported neurological symptoms, including tremor, myoclonus, muscle rigidity, nystagmus/ocular clonus, drowsiness, dysarthria, seizure, and confusion. Some patients complained of dizziness, urinary retention, and jaw stiffness. 

In terms of clinical signs at presentation, one patient had fever, two had bradycardia, and four had tachycardia. Four patients had high blood pressure. No patients had hypotension. The Glasgow Coma Scale score was less than 15 in eight, including two patients with a score less than 9. 

Seizure was reported in one 57-year-old man. He experienced one brief generalized tonic-clonic seizure 4 h after ingestion in conjunction with myoclonus and nystagmus and was intubated for mechanical ventilation. He was also treated with intravenous levetiracetam. His hospital course was complicated by sepsis prior to discharge.

Most patients (94.9%) were admitted to the hospital. The median length of hospital stay was 3 days (range 1–42); four patients were hospitalized for more than 1 week. The median time to clinical improvement was 1 day. The clinical effects of seven patients improved more than 3 days after ingestion. Four patients (10.3%) were admitted to the Intensive care unit. 

The initial laboratory studies are presented in Table 3. Out of thirty patents who had initial serum potassium in their records, thirteen patients (43.3%) had hypokalemia (range of serum potassium: 2–3.4 mEq/L). Ten patients with hypokalemia had nausea/vomiting. Abnormal initial serum sodium was not detected in our patients. Two patients had elevated levels of aspartate aminotransferase (AST) and alanine aminotransferase (ALT) at presentation. One had an AST of 88 IU/L and an ALT of 79 IU/L, the other one who ingested cicada nymphs with alcohol, had an AST of 94 IU/L and an ALT of 138 IU/L. 

The creatine kinase concentration was tested in 11 patients; no patient was diagnosed with rhabdomyolysis. Computed tomography of the head and lumbar puncture were performed in five and two patients, respectively. The results of these investigations were unremarkable.

In 2014, four fungus-infected cicada nymph specimens collected in different areas of Bangkok were sent to the clinical microbiology laboratory of the Mahidol University Department of Pathology. Three were identified as *Ophiocordyceps sobolifera*; the species was not identifiable in the other.

Interestingly, one 47-year-old woman presented with persistent nausea and vomiting, tremor, and myoclonus. Blood and urine specimens collected approximately 1 day after ingestion, were sent for extensive analysis. The predominant peaks were detected in both blood and urine samples using liquid chromatography quadrupole time-of-flight mass spectrometry (LC-QTOF-MS/MS), with the fragmentation pattern relating to ibotenic acid. Muscimol and strychnine were not detected by LC-QTOF-MS/MS. The fungus-infected cicada nymphs she ingested were not examined to determine the fungal species.

Patients with GI symptoms received supportive care including intravenous hydration and antiemetics. Four patients were intubated and underwent mechanical ventilation. These four patients’ ages were 16, 30, 59, and 77 years. Half of them were male. All cooked before eating. The onset of symptoms ranged from 1 h to 5 h after exposure. All patients had nausea/vomiting, tremor, and myoclonus. Their ventilator support lasted for 1, 3, 5, and 31 days. Their hospital stays ranged from 2 days to 42 days. Three patients were admitted for more than 1 week. The 77-year female patient who had the longest length of ventilator support and hospital stay presented to the hospital as a poisoning group member. She developed infections and was diagnosed with suspected diaphragmatic paralysis during hospitalization. Her condition improved approximately 3 weeks after exposure. She was finally discharged after staying in the hospital for 42 days.

Eleven patients received intravenous benzodiazepine (diazepam or midazolam). Intravenous antiepileptic drugs were administered in two patients (phenytoin and levetiracetam, respectively). One patient was diagnosed with coronavirus disease 2019 pneumonia on the day of admission and four developed complications during hospitalization, including sepsis, pneumonia, and urinary tract infection. All patients survived and were eventually discharged.

## 3. Discussion

Thirty-nine cases of human poisoning related to ingestion of fungus-infected cicada nymphs were reported in Thailand from June 2010 to June 2022. All patients involved reported that the infected nymphs were intentionally consumed as a health supplement or herbal treatment; they possibly believed that the infected nymphs were similar to the nymphs infected with *C. sinensis* or *C. militaris,* which are commonly used in TCM to treat disease or enhance immunity [1,2]. In Thailand, there have been reports of poisoning caused by ingestion of plants or animals used as health supplements. [11,12,13]. Based on our data, poisoning occurs after ingestion of fungus-infected nymphs. Moreover, no studies support any health benefits associated with their consumption. As a preventative measure, the general public should be educated that ingestion carries a risk of poisoning. One previous study in the English literature has reported this type of poisoning [2]. The other study was a poison center’s annual report that described three cases in one family with cicada flower poisoning [14]. These patients developed GI symptoms, vertigo, and generalized weakness after consuming golden chan hua, one type of cicada flower [14] Our study helps to clarify the clinical effects of fungus-infected cicada nymph poisoning and suggests that ibotenic acid might be the responsible agent. 

Most poisoning events occurred in May and June, which is the late summer to early rainy season in Thailand [10]. Cicada nymphs may be more susceptible to fungal infection in the high humidity conditions of the rainy season [8]; thus, human poisoning events may be more likely during this period. The clinical effects of poisoning in our study were mostly consistent with those reported in the previous study from Vietnam [2] and the cases described in the poison center’s annual report from Hongkong [14]. The GI and neurological symptoms and signs were the most common manifestations; however, unlike the previous study, not all of our patients experienced GI symptoms. In addition, almost half of our patients experienced myoclonus, which was not reported in the previous study [2]. This might be partially explained by interstudy differences in the interpretation of tremor and myoclonus. Furthermore, salivation, which was reported in all patients of the previous study [2], was not reported in our patients. In our study, seizure was noted only in one patient, so, it was not a common finding of this type of poisoning. In the previous study [2], seizure was reported together with tremor in 75% of cases. However, the authors did not separately state the number of patients who developed seizure. Future large-scale studies are warranted to better clarify the clinical features of poisoning related to the ingestion of fungus-infected cicada nymphs. Almost all patients in our study developed symptoms, only two patients did not have any symptoms after exposure. Therefore, we could not identify the factors associated with developing the poisoning after ingestion.

Ibotenic acid was detected in cicada flower specimens examined in the previous study [2] as well as the blood and urine specimens from one patient in our study. Notably, ibotenic acid was not detected in the unused herb sample, as described in the published poison center’s annual report [14]. Ibotenic acid contributes to neurotoxicity [6,7]. Ibotenic acid and muscimol are major active substances contained in *Amanita muscaria* and *A. pantherine* mushrooms [4,5]. The clinical effects of poisoning by these mushrooms in the central nervous system include agitation, hallucinations, tremor, drowsiness, respiratory depression, and coma [15,16,17,18,19,20]. Myoclonus and ocular clonus/nystagmus are not common with ibotenic acid and muscimol toxicity in humans [15]; however, they were reported in almost half of our patients. Typically, the clinical effects of *A muscaria* and *A. pantherine* poisoning do not last long [15,19,20]. However, in one report, the patient was in a state of coma for 72 h after ingestion of *Amanita muscaria* and stayed in a hospital for 4 days [21]. The other one case report described symptoms persisting for 5 days in a patient who consumed *A. muscaria*; this patient also experienced a seizure-like episode, altered consciousness, and paranoid psychosis [22]. Some of our patients experienced clinical effects that might be longer than those caused by ibotenic-acid- and muscimol-containing mushrooms [15,16,17,18,19,20].

Ibotenic acid is a colorless, crystalline water-soluble substance that is metabolized in vivo by decarboxylation to muscimol [6,7]. Interestingly, in our patient who underwent blood and urine sample analysis, only ibotenic acid was discovered, not muscimol. However, the specimens were collected late after exposure and only one case’s urine was examined, so more case samples should be collected for laboratory analysis to support and confirm this finding. In the previous study from Vietnam, *Ophiocordyceps heteropoda* and ibotenic acid were responsible for the poisoning [2]. By contrast, we identified *Ophiocordyceps sobolifera* in three of four specimens. However, according to the poison center’s annual report [11], the morphological examination of the leftover herb identified *Ophiocordyceps heteropoda* and *Ophiocordyceps sobolifera*, which is consistent with our finding. Previously reported species of fungi infecting cicada nymphs in Thailand include *Polycephalomyces nipponicus*, *Ophiocordyceps longissima*, *Simplicillium obclavatum*, *Metacordyceps chlamydosporia*, and *Ophiocordyceps sobolifera* [9]. Because we only obtained four specimens for fungal culture from 39 patients, we cannot comment further. 

Strychnine poisoning was one of the differential diagnoses in our patients. Strychnine is a strong competitive inhibitor of the postsynaptic glycine receptor, predominantly in the spinal cord [23]. The typical clinical findings of strychnine toxicity include tremor, involuntary generalized muscle contractions, opisthotonos, facial trismus, risus sardonicus, and respiratory muscle spasm [24,25]. According to the sparing of consciousness, patients typically present with ‘awake seizures’ without a postictal phase [23,24,25]. Nevertheless, there was no history of strychnine exposure in every patient, and it was not detected in one patient’s samples. Altogether, our patients should not have experienced strychnine poisoning.

Our findings support the hypothesis that ibotenic acid might contribute to the clinical manifestations of poisoning related to ingestion of fungus-infected cicada nymphs. Nevertheless, further studies are warranted to clarify the fungal species and toxins involved, as well as the underlying mechanisms and pathophysiology. 

Interestingly, hypokalemia occurred at presentation in 43.3% of patients in our study. They mostly had nausea/vomiting. However, our data did not describe the investigations for the pathophysiology of hypokalemia that occurred. Vomiting contributes to hypokalemia via several complex mechanisms [26,27]; however, not all patients had nausea/vomiting. The previous studies referred to did not indicate hypokalemia in their patients [2,14]. Therefore, the mechanism underlying this electrolyte disturbance should be investigated and studied to elucidate this further.

Rhabdomyolysis was not recorded in our patients’ data. Thus, these abnormal neuromuscular activities which occurred as tremor, myoclonus, or muscle rigidity, did not contribute to muscle breakdown and other severe complications.

Although no deaths occurred among the patients in our study, one occurred in the study from Vietnam [2]. Moreover, some patients developed complications and required lengthy hospitalization. Therefore, educating the general public regarding poisoning associated with the ingestion of fungus-infected cicada nymphs would be worthwhile. In addition, extensive study, mainly related to the safety of use as a health supplement, is warranted in the future.

This study has several limitations. First, it was retrospective in design and the data were obtained via telephone follow-up; thus, selection or recall or reporting biases may have been present. Second, as the poisoning is uncommon, there were a small number of patients in this study. Third, the poisoning diagnosis was primarily based on patient reporting of ingestion of fungus-infected cicada nymphs and development of symptoms. No extensive investigations were performed to exclude other possibilities or to confirm the diagnosis of poisoning by ingestion of fungus-infected cicada nymphs. Finally, we could not definitively identify the fungal species or toxins involved in all patients because these analyses are not practically available in Thailand. 

## 4. Conclusions

Human consumption of fungus-infected cicada nymphs may result in poisoning. The GI and neurological symptoms were common in our series of patients and particularly included altered consciousness, tremor, and myoclonus. These symptoms should be examined and observed in people who ingest fungus-infected cicada nymphs. Ibotenic acid was identified and might be the cause. The main treatment is supportive care and appropriate management of complications. We advocate education of the general public to help prevent the incidence of this type of poisoning.

## 5. Materials and Methods

### 5.1. Study Design

This was a retrospective cohort study of patients who ingested fungus-infected cicada nymphs that were referred for consultation to the Ramathibodi Poison Center (RPC) of the Ramathibodi Hospital, Mahidol University from June 2010 to June 2022. Ramathibodi Hospital is an academic tertiary hospital in Bangkok, Thailand. We aimed to examine and describe the clinical characteristics, symptoms, treatments, and clinical outcomes of these patients. Those with incomplete follow-up data were excluded. The study was approved by the institutional ethics committee of the Ramathibodi Hospital Faculty of Medicine, Mahidol University. The requirement for informed consent was waived owing to the retrospective observational design of the study and the anonymized nature of the data. 

### 5.2. Study Site and Patients

The RPC provides information and consultations regarding poisoning to both the general public and healthcare professionals across all of Thailand 24 h a day, 7 days a week. Most calls are from physicians and other health care personnel. Approximately 20,000 to 35,000 calls are handled every year. Follow-up calls are made to record patient progress, provide ongoing management recommendations, and to determine the final outcomes. All cases are recorded in the Toxic Surveillance System database. All records are reviewed and verified by medical toxicologists and senior specialists in poison information.

Patients who had ingested fungus-infected cicada nymphs and were referred to the RPC for consultation during the study period were eligible for study inclusion. Ingestion of fungus-infected cicada nymphs was confirmed by the patient’s or a relative’s description of cicada nymphs or by photographs forwarded to the RPC for review. 

### 5.3. Study Protocol

All patients who met the inclusion criteria were included for analysis. Data regarding patient characteristics, medical history, clinical features, laboratory results, treatment, follow-up details, final diagnosis, and outcome were recorded. Vital sign and Glasgow Coma Scale score abnormalities were defined according to age group [28]. Hyponatremia and hypernatremia were defined as serum sodium <135 and >145 mEq/L, respectively. Hypokalemia and hyperkalemia were determined as serum potassium <3.5 and >5.0 mEq/L, respectively [29]. Elevated levels of aspartate aminotransferase (AST) and alanine aminotransferase (ALT) were determined if the patient’s laboratory results were more than the normal range of each hospital’s laboratory levels. Rhabdomyolysis was defined as a serum creatine kinase concentration >1000 U/L [30] or documentation of it in the patient’s record. 

#### Method for Fungus Culture and Identification of Fungal Species

The cicada samples were analyzed at the microbiology laboratory of the Department of Pathology, Ramathibodi Hospital. Briefly, the surface of each sample was wiped with 70% ethanol to reduce any surface contamination. The samples were then cut, and the internal parts were collected and cultured on Sabouraud dextrose agar and Sabouraud dextrose agar with gentamicin (Becton Dickinson, Franklin Lakes, NJ, USA). The culture media were incubated at 30 °C for up to 30 days. Each sample was observed for fungal growth twice weekly. Identification of any suspected fungus colonies was performed by slide culture and microscopic examination. The organisms were also identified by 18sRNA sequencing.

### 5.4. Statistical Analysis

The data were recorded and analyzed using Excel software (Microsoft Corp., Redmond, WA, USA). Continuous data with a normal distribution are expressed as means with standard deviation; those with a non-normal distribution are expressed as medians with range. Categorical data are expressed as frequencies with percentages. 

## Figures and Tables

**Figure 1 toxins-16-00022-f001:**
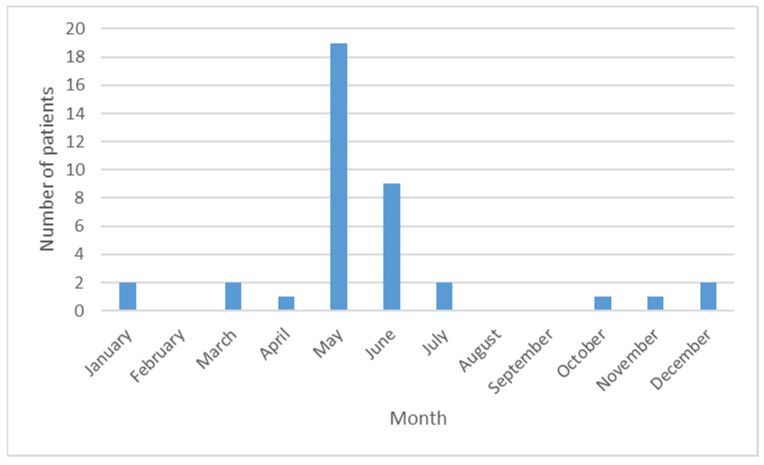
Number of patients with fungus-infected cicada nymph poisoning according to month.

**Table 1 toxins-16-00022-t001:** Patient characteristics.

Characteristic	Number (%)
**Sex**	
Male	21 (53.8%)
Female	18 (46.2%)
**Age (years)**	40.2 ± 15.0
**Region**	
Northeast	17 (43.6%)
Bangkok	13 (33.3%)
South	3 (7.7%)
West	3 (7.7%)
North	2 (5.1%)
Central	1 (2.6%)
**Season**	
Summer	22 (56.4%)
Rainy season	11 (28.2%)
Winter	6 (15.4%)
**Cooking method**	
Frying	25 (64.1%)
Boiling	5 (12.8%)
None/raw	3 (7.7%)
No data	6 (15.4%)

Data shown are numbers (%) or means ± standard deviation.

**Table 2 toxins-16-00022-t002:** Clinical manifestations at presentation.

Clinical Manifestation	Number (%)
**Gastrointestinal symptoms ***	
Nausea/vomiting	31 (79.5%)
Diarrhea	5 (12.8%)
Abdominal pain	1 (2.6%)
**Neurological symptoms ***	
Tremor	21 (53.8%)
Myoclonus	20 (51.3%)
Muscle rigidity	13 (33.3%)
Nystagmus/ocular clonus	12 (30.8%)
Drowsiness	8 (20.5%)
Fasciculation	5 (12.8%)
Dysarthria	4 (10.3%)
Confusion	3 (7.7%)
Muscle weakness	2 (5.1%)
Seizure	1 (2.6%)
**Others**	
Dizziness	9 (23.1%)
Urinary retention	3 (7.7%)
Jaw stiffness	2 (5.1%)
Diaphoresis	2 (5.1%)

* Some patients had > 1 manifestation.

**Table 3 toxins-16-00022-t003:** Laboratory results of all patients at presentation.

Laboratory Results (Number of Patients with Data Available)	Value
Hematocrit (%); mean ± SD (*n* = 18)	37.4 ± 4.7
White blood cells (per microliter); mean ± SD (*n* = 21)	11,263.3 ± 5526.0
Platelets (per microliter); mean ± SD (*n* = 17)	271,176.5 ± 59,611.7
Serum sodium (mEq/L); mean ± SD (*n* = 28)	141.6 ± 2.8
Serum potassium (mEq/L); mean ± SD (*n* = 30)	3.5 ± 0.5
Serum chloride (mEq/L); mean ± SD (*n* = 25)	103.9 ± 2.7
Serum bicarbonate (mEq/L); mean ± SD (*n* = 28)	24.1 ± 3.1
Serum blood urea nitrogen (mg/dL); mean ± SD (*n* = 21)	11.6 ± 4.4
Serum creatinine (mg/dL); median (min-max) (*n* = 25)	0.78 ± 0.17
Serum aspartate aminotransferase (IU/L); median (min–max) (*n* = 16)	22.5 (14–94)
Serum alanine transaminase (IU/L); median (min–max) (*n* = 16)	32 (11–138)

## Data Availability

The data presented in this study are available on request from the corresponding author. The data are not publicly available because of patient privacy concerns.

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
