# Peer review of "Poisoning from Ingestion of Fungus-Infected Cicada Nymphs: Characteristics and Clinical Outcomes of Patients in Thailand"

_toxins, 2023, doi:10.3390/toxins16010022_

Round 1

Reviewer 1 Report

Comments and Suggestions for Authors

The authors did a retrospective cohort study of patients who ingested fungus-infected cicada nymphs and added to the knowledge base of poisoning associated with ingestion of fungus-infected cicada nymphs.

Some of my specific comments are below.

  1. Incomplete sentence at line 64, and 222.
  2. Missing . at line 25.
  3. Line 66, the minus sign seems to be wrong.
  4. Table 2,   Nystagmus/ocular clonus and drowsiness have no percentage associated with them.
  5. At line 234, "Continuous data with a normal distribution are expressed as means with standard deviation; that with a non-normal distribution are expressed as medians with range." Which test is used to determine normality? 
Comments on the Quality of English Language

There are some incomplete sentences and missing punctuations as noted above.

Reviewer 2 Report

Comments and Suggestions for Authors

Dear Authors, I have read your manuscript with interest.

The current manuscript titled: "Poisoning from Ingestion of Fungus-Infected Cicada Nymphs: Characteristics and Clinical Outcomes of Patients in Thailand" represents an important analysis of evolving field of Toxicology.

In my opinion, these are the adjustments which should be made to increase the value of your manuscript:

1.       In Introduction chapter, please, describe in detail the physiopathological mechanisms causing intoxication with the presented toxin.

2.       In Table 1, add more detailed patients’ baseline characteristics.

3.       In the Discussion section, there is not enough comparative information with other studies. Please, add.

4.       It is recommended to study and consider this recent holistic article about acute poisoning https://doi.org/10.1155/2021/4696156.

5.       Please, add future perspectives.

6.       The Conclusions section, please, highlight the practical implications of this study and its relevance to real clinical practice.

7.       The manuscript contains some punctuation errors, please revise the text.

Comments on the Quality of English Language

Minor editing of English language required.

Reviewer 3 Report

Comments and Suggestions for Authors

1. This is a retrospective cohort study on poisoning caused by ingestion of fungus-infected cicada nymphs in Thailand during 2010-2022. The study content has some novelty and can provide more information on the clinical features and outcomes of this type of poisoning.

2. The study design is reasonable, but as a retrospective study, there is a possibility of case selection bias and recall bias. It is recommended that the authors point out this limitation in the discussion section.

3. The sample size is small (39 cases) from a single hospital in one region. The authors need to mention in the discussion that the representativeness of the results may be limited. 

4. The collection of case data is incomplete, especially in terms of etiological examinations. Only 4 cases underwent fungal culture and identification. It is recommended that the authors discuss the necessity of expanding the sample size and improving etiological examinations.

5. No intra-group controlled studies were conducted, so it is uncertain whether the clinical manifestations were significantly different from the healthy control group.

6. The discussion section somewhat exaggerates the significance of the study, and a more cautious expression of the research conclusions is needed.

7. The format of literature citations needs to be unified to meet the requirements of the target journal.

8. Overall, the research topic has some value and significance, but the research methods and discussion of results need to be more prudent, requiring further improvement by the authors. It is hoped that the above comments can provide some reference for the authors to improve the paper.

Reviewer 4 Report

Comments and Suggestions for Authors

This is a very interesting summative case series report.

53. I am curious how 39 patients were detected from 23 telephonic consultations - this implies multiple cases per consultation. Please clarify.

65. There seems to be some missing words. It states: "Most cases were reported in"...I am assuming this is "May (48.7%)."

Figure 1. I'm not sure that the figure is necessary. I would suggest adding months with counts and percentages to Table 1. 

You may be interested in reading "Case 6 - cicade flower poisoning"in this publication: https://journals.sagepub.com/doi/10.1177/10249079221127611?icid=int.sj-abstract.similar-articles.7#bibr32-10249079221127611

Round 2

Reviewer 2 Report

Comments and Suggestions for Authors

I agree with the changes made, which significantly improve the quality of the manuscript.

Reviewer 3 Report

Comments and Suggestions for Authors

I am very glad to receive another letter from the author, the article has been modified with tight logic and general expression, and I suggest that the magazine receive it.

Comments on the Quality of English Language

The English level expression of the article is reasonable